# Volatile Organic Compounds, Oxidative and Sensory Patterns of Vacuum Aged Foal Meat

**DOI:** 10.3390/ani10091495

**Published:** 2020-08-24

**Authors:** Alessandra Tateo, Aristide Maggiolino, Ruben Domínguez, José Manuel Lorenzo, Francesca Rita Dinardo, Edmondo Ceci, Rosaria Marino, Antonella della Malva, Andrea Bragaglio, Pasquale De Palo

**Affiliations:** 1Department of Veterinary Medicine, University of Bari A. Moro, 70010 Valenzano, BA, Italy; Alessandra.tateo@uniba.it (A.T.); francesca.dinardo@uniba.it (F.R.D.); edmondo.ceci@uniba.it (E.C.); andrea.bragaglio@uniba.it (A.B.); pasquale.depalo@uniba.it (P.D.P.); 2Centro Tecnológico de la Carne de Galicia, Parque Tecnológico de Galicia, 32900 Ourense, Spain; rubendominguez@ceteca.net (R.D.); jmlorenzo@ceteca.net (J.M.L.); 3Area Tecnologia de los alimentos, Facultad ciencias de Oruesnse, Universidad de Vigo, 32004 Ourense, Spain; 4Department of Agricultural Food and Environmental Sciences, University of Foggia, 71121 Foggia, Italy; rosaria.marino@unifg.it (R.M.); antonella.dellamalva@unifg.it (A.d.M.)

**Keywords:** foal meat, volatile organic compounds, vacuum packaging, oxidative status, sensory evaluation

## Abstract

**Simple Summary:**

Aging is the way that leads the transformation of muscle in meat, usually improving sensorial and textural properties. Aging in horse meat was recently investigated, although it needs to be deepened for a better application of this kind of technique. Vacuum aging is a spread aging technique, mainly because it represents a cheaper method, as it does not need aging rooms, presents less weight loss than dry aging, and it is safer because primal cuts are protected by air contaminations. The present paper characterizes volatile compounds, oxidative status, and the sensory profile of foal meat during vacuum aging for 14 days. Vacuum aging seems to delay lipid oxidation, reducing oxidative products and volatile compound production derived from lipid oxidation.

**Abstract:**

The study aimed to evaluate the effect of 14-day vacuum aging on the volatile compounds (VOC) profile, oxidative profile, antioxidant enzymes activity, and sensory evaluation in the Longissimusthoracis muscle of foal meat under vacuum aging. Longissimusthoracis (LT) was sampled in 20 mm thick slices, vacuum packed, and stored at 4 °C. Samples were randomly assigned to different aging times (1, 6, 9, 14 days after slaughtering). VOCs, thiobarbituric acid reactive substances (TBARs), hydroperoxides, carbonyl proteins, superoxide dismutase, catalase, and glutathione peroxidase were analyzed, and a sensory test was performed. A nested one-way analysis of variance (ANOVA) was performed for aging time as an independent variable. Significance was set at *p* < 0.05. The main VOCs originating from cooked steaks were aldehydes, (from 47.18% to 58.81% of the total volatile compounds), followed by hydrocarbons (from 9.32% and 31.99%). TBARs and hydroperoxides did not show variations due to aging (*p* > 0.05), instead, protein carbonyls showed higher values at the 14th day (*p* < 0.01). Catalase, superoxide dismutase, and glutathione peroxidase showed increasing values during aging time (*p* < 0.01). Vacuum aging slowed down lipid oxidation, and protein oxidation was shown to be present. However, the best vacuum aging duration is in the range of 6–9 days from slaughtering, with an improvement of sensory evaluation.

## 1. Introduction

In order to improve eating quality (tenderness and juiciness), meat is normally stored for a period [1], this is in order to lead the muscle conversion to meat, and aging, which depend on several factors, such as species, muscle type, packaging conditions, slaughter age, sex, intramuscular fat content, breed, etc. In addition, other sensorial patterns (taste, odor, and flavor) could be affected by aging period [2]. Moreover, meat flavor can be defined as those attributes in meat that affect the taste, which is the result of a combination of taste compounds registered on the tongue (sweet, salty, bitter, sour, and “umami”), and odor or aroma detected in the nose.

During aging, a release of peptides, free amino acids, and free fatty acids is expected due to lipid degradation and proteolytic breakdown of the proteins, which lead to increase of flavor compounds [3]. In addition, the volatile compounds generated during cooking depend both on the content of precursors in the raw meat (free amino acids, peptides, reducing sugars, vitamins, and lipids) and on the temperature reached by the sample meat during the cooking process. In this regard, some authors studied the effect of four different cooking methods (roasting, grilling, microwaving, and frying) on the volatile profile of foal meat [4], and they reported that roasted steaks showed the highest volatile content, indicating that increasing cooking temperature increases the formation of volatile compounds.

On the other hand, in comparison to other red meat, horse meat is characterized by low intramuscular fat and cholesterol content, and a great amount of iron [5]. In addition, this meat presents a high proportion of unsaturated fatty acids in relation to saturated, and has a great amount of α-linolenic fatty acid [6]. Due to these nutritional properties, horse meat consumption is slowly increasing in several western European countries based on claims that it could be a healthier red meat alternative [7].

For consumer satisfaction, aroma compounds released from meat are very important. Meat aroma is really complicated due to the great variety and variability of compounds that bring a combination of many different flavor notes, which, together, generate the meat aroma [8]. In this regard, the knowledge of these volatile compounds responsible for aroma perception, and how their concentrations change during foal meat aging, is of interest toward achieving a high meat quality.

Previous studies [2,9,10] showed that refrigerated vacuum aging for a period of 7–30 days increased the flavor score of cooked meat. This effect has also recently been confirmed in foal meat, although the overall consumer acceptance of vacuum aged horse meat was based more on textural improvement than on flavor/taste increase [11].

The current work aimed to contribute to the overall assessment of the effect of vacuum aging for 14 days on the volatile compounds profile, the oxidative status, and the sensory evaluation of horse meat.

## 2. Materials and Methods

### 2.1. Animal and Sampling Methods

The trial involved 10 Italian Heavy Draft Horse (IHDH) male foals. All animals were born and fattened until 12 months of age in the same farm. They were individually reared and had the same feed and ratio.

At the age of 365 ± 7 days, each foal was transported and slaughtered at a European Community-approved abattoir, according to European Community laws on Animal Welfare in transport (1/2005EC) and the European Community regulation on Animal Welfare for slaughter of commercial animals (1099/2009EC), located 35 km from the farm, as described by Maggiolino et al. [12]. The journey time was 32 min (±6, standard deviation; SD). Foals were stunned (by captive bolt gun), exsanguinated, and dressed according to local commercial dressing-out procedures at the abattoir [13]. No electrical stimulation was used. After slaughtering procedures, carcasses were chilled at 4 °C in a chilling room for 24 h. Afterwards, the *Longissimus thoracis* (LT) muscle was sampled from the right half-carcass, between the fourth and the ninth rib, of each foal. Each muscle was divided into four sections, stored at 2 °C under vacuum packaging, and randomly assigned to one of the different aging times (1, 6, 9 and 14 days of aging); cranial and caudal sections were randomized across aging time. All samples were vacuum packaged with Besser Vacuum^®^ film (Besser Vacuum, Dignano, Udine, Italy) characterized by 65 µm thickness, 63 g/m^2^ of weight, ≤ 65 cm^3^/m^2^ × day × bar of oxygen permeability at 23 °C and 0% of relative humidity, and ≤ 3.5 g/m^2^ × day of water vapor permeability at 23 °C and 85% of relative humidity.Ten samples were obtained and analyzed at each aging time.

### 2.2. Chemical Composition

Chemical composition was performed exclusively on samples at day 1. Moisture [14], protein [15], intramuscular fat [16], and ash [17] content were calculated with the International Organizartion for Standardization (ISO) methods. Samples were analyzed in triplicate.

### 2.3. Volatile Compound Analysis

At each aging time, two 5 g samples of each foal steak were grilled at 130–150 °C for 5 min on each surface, using an electrical griddle (Delonghi, Mod. CG660, Treviso, Italy). A heating treatment was considered complete when all the steaks reached an internal temperature of 70 °C measured with a copper constantan fine-wire thermocouple fixed in the geometrical center of the sample (Model 5SCTT-T-30-36; Omega Engineering Inc., Norwalk, CT, USA). After cooking and cooling, the samples were minced using a commercial grinder (Moulinex/Swan Holding Ltd., Birmingham, England), vacuum-packed, and stored at −30 °C for no longer than two weeks until analysis.

#### 2.3.1. Solid-Phase Micro Extraction (SPME)

The SPME tool from autosampler was loaded with a fused-silica fiber (10 mm length) coated with a 50/30 mm thickness Divinylbenzene/Carboxen/Polydimethylsiloxane (of DVB/CAR/PDMS) (Supelco, Bellefonte, PA, USA). Before the analysis, the fiber was conditioned by heating in a SPME Fiber Conditioning Station at 270 °C for 30 min. For headspace SPME (HS-SPME) extraction, 1 ± 0.02 g of each sample was weighed in a 20 mL vial (Agilent Technologies, Santa Clara, CA, USA) and subsequently screw-capped with a laminated Teflon-rubber disc. The extractions were carried out at 37 °C for 30 min, after equilibration of the samples for 15 min at the same temperature, ensuring a homogeneous temperature for sample and headspace.

#### 2.3.2. Chromatographic Conditions

After the extraction procedure, the fiber was transferred to the injection port of the gas chromatograph–mass spectrometer (GC–MS) system (7890B GC-System; Agilent Technologies, Santa Clara, CA, USA and a mass selective detector 5977B MSD; Agilent Technologies, Santa Clara, CA, USA). The column used for volatile separation was a DB-624 capillary column (30 m, 250 µm i.d., 1.4 μm film thickness; J&W Scientific, Folsom, CA, USA). The chromatographic conditions and mass spectrometer parameters were previously described [18].

#### 2.3.3. Data Processing

After chromatographic analysis, all data were analyzed with the software MassHunter Quantitative Analysis B.07.01 (Agilent Technologies, Santa Clara, CA, USA). A new method from acquired scan data with library search was created. The integration was done with Agile2 (Agilent Technologies, Santa Clara, CA, USA) algorithm, while peak detection was done with deconvolution. Compounds were identified by comparing their mass spectra with those contained in the National Institute of Standards and Technology (NIST) 14 library (Gaithersburg, MD, USA). The compounds were considered as correctly identified when their spectra presented a library match factor > 85. After integration, peak detection, and identification of each compound, the Extraction Ion Chromatogram (EIC) from the quantifier ion was obtained from each peak. The results were expressed as area units of the EIC × 104 per gram of sample (area units (AU)-EIC × 10^4^/g of sample).

### 2.4. Thiobarbituric Acid Reactive Substances (TBARS), Hydroperoxides, and Protein Carbonyls Analyses

Two raw minced samples (5 g) for each foal steak at each aging time were placed in a 50-mL test tube and homogenized with 15 mL deionized distilled water (DDW). One ml of homogenate was moved to a glass tube for the TBARS determination and 0.05 mL of butylated hydroxytoluene (7.2% in ethanol) was added along with 1950 mL of thiobarbituric acid (TBA)/trichloroacetic acid (TCA)/HCl (0.375% TBA, 15% TCA and 0.25 N HCl). The obtained solution was shaken and incubated at 90 °C for 15 min in a thermostatic bath. After, samples were cooled at room temperature (15 °C–30 °C) and then centrifuged at 2000× *g* for 15 min. Supernatant absorbance at 531 nm was measured against a blank containing 2 mL of TBA/TCA/HCl solution in 1 mL of distilled water. The TBARS were calculated comparing with a standard curve constructed with 1,1,3,3-tetramethoxypropane, and the concentration of lipid oxidation was expressed as milligrams of malondialdehyde (MDA) per kg of meat [19].

For hydroperoxides determination, 2 mL of homogenate (previously prepared for the TBARS determination) were added with 4 mL of CH3OH and 2 mL of CHCl3. The samples were vortexed for 30 s and were added with 2 mL of CHCl3 and 1.6 mL of 0.9% NaCl. The samples were shaken for 1 min and then centrifuged at 3500× *g* for 10 min at 4 °C. Two milliliters of lipid extract were sampled from the lower chloroform phase and processed with 1 mL of CH3COOH/CHCl3 and 50 μL of potassium iodide (KI) (1.2 g/L mL distilled water). Samples were stored for 5 min in a dark room and added with 3 mL of 0.5% of CH3COOCd, and then vortexed and centrifuged at 4500× *g* for 10 min at 40 °C. Absorbance at 353 nm was measured against a blank tube in which meat homogenate was replaced by 2 mL of distilled water [20]. Results were expressed in micromoles per gram according to Buege and Aust [19].

Two raw meat samples (2 g) for each foal steak at each aging time were homogenized in 20 mL of 0.15 M KCl for 2 min. Two aliquots of homogenate (50 μL each) were added with 1 mL 10% TCA and then centrifuged at 1200× *g* for 3 min at 4 °C to measure protein oxidation. The first aliquot was used as a standard and added with 1 mL of 2 M HCl solution. The second aliquot was added with 1 mL of 2 M HCl containing 10 mM 2,4-dinitrophenyl hydrazine (DNPH). Samples were incubated for 1 h at room temperature (15 °C to 30 °C) and shaken every 20 min, and then 1 mL of 10% TCA was added. The samples were vortexed for 30 s and centrifuged 3 times at 1200× *g* for 3 min at 4 °C and the supernatant removed. Care was taken not to disrupt the pellet. The pellet was washed with 1 mL of ethanol:ethyl acetate (1:1), shaken, and centrifuged 3 times at 1200× *g* for 3 min at 4 °C and the supernatant removed. The pellet was then dissolved in 1 mL 20 mM sodium phosphate 6 M guanidine hydrochloride buffer. Samples were then shaken and centrifuged at 1200× *g* for 3 min at 4 °C. Carbonyl concentration was calculated on the DNPH treated sample at 360 nm with a Beckman Coulter DU800 (Beckman Instruments Inc., Brea, CA, USA) and expressed as nanomoles carbonyl per milligrams protein. Protein concentration was calculated according to Biuret assay [21,22].

### 2.5. Superoxide Dismutase, Catalase, and Glutathione Peroxidase Activity Evaluation

Two samples of about 400 mg of raw meat for each foal steak at each aging time were homogenized in a tissue homogenizer 4 mL saline at 4 °C. The homogenate was centrifuged at 4 °C for 20 min at 7000× *g* and supernatant was collected to determine the antioxidant enzyme activities. Plasma was analyzed as it was.

Superoxide dismutase (SOD, EC 1.15.1.1) was evaluated according the method of Misra [23]. The activity was determined considering its ability to inhibit the epinephrine autoxidation. Stimulation of epinephrine autoxidation by traces of heavy metals present as contaminants in the reagents or by the other metals under investigation was prevented by adding 10–4 M ethylenediaminetetraacetic acid (EDTA) in the buffer to chelate these ions. One unit of SOD is defined as the amount of enzyme required to inhibit the rate of epinephrine autoxidation by 50%. The enzyme activity was expressed as U/mg protein.

Catalase (CAT, EC 1.11.1.6) activity was measured by the method of [24], by following the decrease in absorbance of H_2_O_2_ at 240 nm (e =40 M^−1^ cm^−1^). One unit of enzyme activity is defined as the amount of enzyme required to degrade 1 micromole of H_2_O_2_ in 1 min and is expressed as U/mg protein.

Glutathione peroxidase (GPx, EC1.11.1.9.) activity was measured by method of [25]. The reaction measured the rate of reduced glutathione(GSH) oxidation by tert-Butyl hydroperoxide, catalyzed by GPx. GSH was maintained at constant concentration by the addition of exogenous glutathione reductase(GR) and NADPH, which converted the oxidized glutathione (GSSG) to GSH. The rate of GSSG formation was then measured by the change in the absorbance of nicotinamide adenine dinucleotide phosphate reductase (NADPH) at 340 nm. (e = 6.2 mM^−1^ cm^−1^) and activity expressed as nanomoles of NADPH oxidized/min/mg protein.

### 2.6. Sensory Analysis

The sensory analysis was conducted throughout an 8-person trained taste panel. They were selected for their sensory acuity using the methods outlined in American Meat Science Association [26]. At each aging time, samples were unpackaged and, after sampling meat for VOC’s analysis, enzymes, and oxidative profile, they were grilled as previously described for VOCs. Fat and connective tissue were trimmed when visible, and the muscle cut into about 2 cm^3^ blocks. They were wrapped in pre-labelled foils and placed in a heated incubator until test. Samples were tasted in an order based on the designs outlined by [27] for balancing carryover effects between samples. Each panelist received 20 samples each test day (two for each animal), subdivided in four different sitting sessions (six or four samples per session), randomized by the sensory panel software, in a different order for each panelist. Tested samples were scored on a 1–10 point scale for tenderness (1 = extremely tough to 10 = extremely tender), juiciness (1 = extremely dry to 10 = extremely juicy), overall licking (1 = extremely disliking to 10 = extremely licking), sweetness, unpleasant taste, meaty odor, and unpleasant odor, (1 = extremely weak to 10 = extremely strong).

### 2.7. Statistical Analysis

All data were previously tested for normal distribution and variance homogeneity by the Shapiro–Wilk test. For the statistical analysis a nested one-way analysis of variance (ANOVA) using the IBM SPSS Statistics 24.0 program (IBM Corporation, Somers, NY, USA) was performed for all aging time (1, 6, 9, and 14 days), where this parameter was set as independent variable. The mean values and standard error of the means (SEM) were calculated. When a significant effect (*p* < 0.05) was detected, means were compared using the Tukey’stest for repeated measures for differences between aging times.

## 3. Results

The chemical composition of foal meat showed mean values of 73.32 ± 0.73%, 22.63 ± 0.85%, 2.38 ± 0.57% and 1.45 ± 0.18% for moisture, protein, intramuscular fat, and ash content, respectively, (data not shown).

The effect of storage time on aldehydes and furans compounds of LT muscle of foal meat aged up to 14 days is shown in Table 1. Heptanal, nonanal, and 2-Hexenal, (E) showed highest values at nine aging days (*p* < 0.05). Furan, 2-ethyl-, Furan, 2-pentyl-, 2-n-Butyl furan, and total furans at day 9 showed the highest values (*p* < 0.05). Moreover, at day 14, Furan, 2-ethyl- and Furan, 2-pentyl- registered higher values than days 1 and 6 (*p* < 0.05). Total furans at 14 days were higher than 6 days (*p* < 0.05).

The effect of storage time on hydrocarbons and aromatic hydrocarbons compounds of LT muscle of foal meat aged up to 14 days is shown in Table 2. Pentane, isopropylcyclobutane, and 1,2 dimethyl cyclohexane were significantly (*p* < 0.05) affected by aging. These volatile compounds presented a similar trend, increasing from days 1 to 9 of aging (*p* < 0.05) and then only cyclohexane decreased (*p* < 0.05) until the end of storage period. Benzene and 3-carene (*p* < 0.05) were significantly affected by the storage period, presenting the highest values after 9 and 6 days of aging, respectively. Total aromatic hydrocarbons increased until 6 aging days and then decreased (*p* < 0.05).

The effect of storage time on ketones and alcohols compounds of LT muscle of foal meat aged up to 14 days is shown in Table 3. The 3-Heptanone showed highest values at 6 and 9 days of aging (*p* < 0.05); at 14 days, showed higher values than the first day (*p* < 0.05). The 2-Heptanone, 4-Hexen-3-one, 5-methyl- and 3-Octen-2-one showed the highest values at 9 aging days (*p* < 0.05). The 1-Octen-3-ol showed, at 9 days, the highest values (*p* < 0.05), and at 14 days higher values than 1 and 6 aging days (*p* < 0.05). Total alcohols increased with aging, showing highest values at 9 aging days (*p* < 0.05).

The effect of storage time on carboxylic acids, total nitrogen compounds, and total sulfur compounds of LT muscle of foal meat aged up to 14 days is shown in Table 4. Butanoic acid showed highest values at 6 aging days (*p* < 0.05), with the lowest values the 1st aging day (*p* < 0.05). Similarly, total carboxylic acids increased at 6 days (*p* < 0.05) and decreased at 14 days, showing higher values than the first aging day (*p* < 0.05).Pyrazine, methyl- showed a decreasing trend with lowest values at 9 aging days (*p* < 0.05). Pyrazine, trimethyl, and total nitrogen compounds decreased constantly until 9 aging days (*p* < 0.05), and at 14 aging days increased again (*p* < 0.05).

Oxidative profile and antioxidant enzymes activity are reported in Table 5. Protein carbonyls increased at 14 days (*p* < 0.01). Superoxide dismutase, catalase, and glutathione peroxidase increased constantly during aging (*p* < 0.01).

Sensory evaluation results are reported in Table 6. Juiciness at 9 and 14 aging days was lower than 6 aging days (*p* < 0.05). Meaty odor and overall liking increased with aging, showing highest values at 9 and 14 aging days (*p* < 0.05).

## 4. Discussion

A total of 117 volatile compounds were identified in the headspace of cooked foal meat samples during the storage time. These volatile compounds were classified according their chemical family as follows: 50 hydrocarbons, 6 aromatic hydrocarbons, 17 aldehydes, 15 ketones, 6 furans, 13 alcohols, 3 carboxylic acids, 5 nitrogen compounds, and 2 sulfur compounds. The overall total compounds detected are almost the same previously recorded in Semimembranosus samples of horse meat [12]. These volatile compounds are generated from lipid degradation (such as aldehydes, alcohols, ketones, and hydrocarbons) or from Maillard reaction (such as sulfur and nitrogen compounds). The total volatile compounds remained constant for the first 6 days (about 35,000 AU × 10^3^/g sample), suffering a significant increase on the day 9 (66,531 AU × 10^3^/g sample) and then they decreased until 45,251 AU × 10^3^/g sample at the end of storage period. This finding is in agreement with data previously reported [28], which recorded the amount of many volatile compounds of cooked lamb meat significantly decreased after 6 days of aging. In addition, some authors also observed that the total volatile compounds isolated from cooked beef increased during aging between 2 and 30 days of storage [10].

The main volatile compounds originated from cooked foal steaks were the aldehydes, ranging from 47.18% to 58.81% of the total volatile compounds, followed by hydrocarbons (between 9.32% and 31.99% of the total volatile compounds). This outcome is in agreement with previously described results [29], where it was highlighted that the most important volatile compounds identified in the headspace of the foal steaks were the aldehydes (ranging from 71% to 87% of the total volatile compounds) followed by linear hydrocarbons (varying between 5% and 14% of the total volatile compounds). In addition, some authors [9] also observed that aldehydes were the main volatile compounds of beef samples aged for 7 and 14 days. Comparing results obtained on the Longissimus thoracis (LT) muscle with the ones recorded in Semimembranosus (SM) muscles [12], the latter showed a lower percentage of aldehydes as well as a lower variability in hydrocarbons percentage. This allow us to highlight that the different chemical and textural properties of each cut [30] could affect the production of volatile compounds both from a quantitative and a qualitative point of view.

On the other hand, out of 117 volatile compounds identified, only 19 compounds including 3 hydrocarbons, 5 aromatic hydrocarbons, 3 aldehydes, 4 ketones, and 2 nitrogen compounds were significantly affected by aging time. According to some authors [10], the lack of significant differences on the volatile compounds among storage times is due to a high variability in the raw data due to the presence of precursors, the complexity in the formation of the volatile compounds, and the interactions that could take place [2]. Although similar chemical composition, and similar protein and intramuscular fat concentration, we cannot exclude differences in fatty acid composition and behavior during vacuum storage that can affect the volatile compounds formation.

Within aldehydes, hexanal, which imparts pleasant fatty green flavor of cooked meat [31], was the most abundant volatile compound, ranging from 80.02% to 86.06% of the total aldehydes in cooked foal meat steaks, followed by pentanal (varying between 5.02% and 6.10% of the total aldehydes). These results agree with what reported on SM muscles of horse meat under the same storage conditions [12]. Both these aldehydes increased during the aging period, but not significantly, due to the variability of the data. This trend is consistent with what was previously reported on SM muscle [12]. According to some authors [32], rancid off-flavor of beef meat was correlated with pentanal and with 2-pentyl furan (metallic, green, earthy, beany) but not with hexanal (grassy, fatty). This result is in agreement with data reported by other authors, who found that hexanal was the main volatile compound isolated from foal [4,12], pork [33], beef [34], and lamb [35] meat. However, some authors [9] found that benzaldehyde was the most abundant in LD and Semitendinosus (ST) muscles from beef aged for 28 days. In our study, benzaldehyde increased 1.98 folds from day 1 to 9, but not significantly. This outcome is consistent with data reported by other authors [10,34] who observed that the level of benzaldehyde increased as the aging period increased. This volatile compound is generated from the Strecker degradation of phenylalanine, in presence of dicarbonyl compounds [36].

Heptanal and nonanal showed a significant increased from day 1 to 9 of storage time and then they decreased. This is consistent with data reported by other authors [10] who found that octanal, nonanal, 2-heptenal, and 2-octenal increased between 1.7 to 3.7 fold from day 2 to 30 of refrigerated period, but not significantly due to the previously stated reasons. The high temperature used for cooking samples may have amplified the oxidation of unsaturated fatty acids and increased the amount of heptanal and nonanal [37]. In our study, the lipid oxidation seems to progress slightly, probably due to the vacuum packaging and/or to the short aging time. The oxidative profile, reported in Table 5, showed that TBARs and hydroperoxides were not affected by aging time, remaining constant during the entire period. A similar trend was previously reported [38] recording that the aging period of 0–2 weeks presented relatively little changes, whereas great variations were observed for the aging period of 2–4 weeks. In addition, some authors [10] noticed that lipid oxidation would be relatively depressed up to 3 weeks under vacuum conditions.

The trends of aldehydes, as well as the significance levels, seem to confirm what was previously reported on SM samples of horse meat under the same conditions [12], highlighting that the chemical pathways that produce these compounds seem to not be affected by muscle. This aspect can be due, also, to the low permeability of the film.

Statistical analysis showed that total furans were significantly affected by aging period, showing the highest values after 9 days of storage time. Furans are normally associated to heat and they are generated from amino acid [9]. According to some authors [39], the concentration of free amino acid and sugars increases during aging time. This study shows that the significant increases of some furans during aging time could be explained by the increase of free amino acids during the storage time and high cooking temperatures, which lead to Maillard reactions. The protein oxidation seems to be affected by aging, showing an increasing trend during aging and higher protein carbonyls values at 14 aging days compared to previous days. These results agree with the slight increase of compounds linked to the greater availability of free amino acids during aging time. It is not clear whether lipid oxidation initiates protein oxidation or vice versa, or even if the two oxidation processes are coupled [20,40], but it is clear that vacuum aging of horse meat showed how these processes go differently, both for the kind and intensity of oxidation, compared to what was reported by other authors in other species’ meat under different packaging methods [20,22,41,42,43]. Findings about furans are in agreement with other reported outcomes [2], where it was described that furans increased with aging in beef steaks aged for 15 days under vacuum conditions and displayed in modified atmosphere packaging during 9 days. However, other authors [10,33,34] did not observe changes in furans during the storage time. The outcomes on the furans are consistent with what previously reported on the SM muscles under the same conditions in terms of trends, although we found statistical significance only on LT muscle.

Regarding hydrocarbons, out of 50 volatile compounds identified, only three volatile compounds (pentane, isopropylcyclobutane and 1,2 dimethyl cyclohexane) were significantly affected by aging. Among this chemical family, 2,2,4,6,6-pentanethyl heptane was the most abundant (between 1650 and 2729 AU × 10^3^/g sample) followed by dodecane (ranged from 430 to 809 AU × 10^3^/g sample) and n-hexane (varied between 130 and 699 AU × 10^3^/g sample). Within aromatic hydrocarbons, 1,3-dimethyl benzene was the main volatile compound, ranging from 286 to 522 AU × 10^3^/g sample without changes during the aging time. However, benzene and 3-carene were significantly affected by the storage period. This finding disagrees with data reported by Domínguez et al. [44], who noticed that toluene was the most abundant followed by o-xylene and p-xylene in cooked foal steaks, although they are consistent with outcomes shown by Maggiolino, Lorenzo, Marino, Della Malva, Centoducati, and De Palo [12]. Toluene can be generated from pyrolyzing free tyrosine or it came from unsaturated hydrocarbons derived from auto- and thermal-oxidative fatty acids [45]. In our study, toluene did not change during the aging time, indicating that lipid oxidation progressed slightly due to the samples that were vacuum packaged and the short storage period. A similar trend was noticed by other authors [34], who did not find any significant difference in the toluene content between 1 and 21 days of storage.

Total ketones were also significantly affected by aging period, showing the highest values after 9 days of storage time. Ketones, especially 2-ketones, have a great impact on the flavor of meat and meat products and they have a peculiar aroma, such as spicy, butter, cheese notes, and ethereal [9]. This result is in agreement with those previously reported [2] where an increase in the amount of ketones with increased storage time was highlighted. In addition, 2-heptanone has been proposed as a candidate marker for the aroma differences between less or more oxidized cooked beef [46].

Regarding alcohols, 1-hexanol was the most abundant volatile compound, ranging from 865 to 10,872 AU × 10^3^/g sample followed by 1-pentanol (from 603 to 3253 AU × 10^3^/g sample) and 1-octen-3-ol (between 153 and 757 AU × 10^3^/g sample). This outcome is in agreement with data previously reported [2], where authors observed that 1-hexanol increased during the aging time due to the growth of natural spoilage bacteria [47], giving ethereal, fruity, fuel oil, pungent, alcoholic, and sweet with green top notes [48]. In addition, other authors [10,34] also found that 1-hexanol and 1-octen-3-ol increased with storage time.

Statistical analysis showed that total carboxyl acids were significantly affected by aging period. According to some authors [48], butanoic, hexanoic, and nonanoic acids come from the spoiled raw meat, impairing cheesy, gammy, fatty, and dairy odor. These results disagree with those reported by some authors [49], who did not find hexanoic acid in raw beef meat under vacuum conditions. In addition, other papers [50] observed that hexanoic acid was only detected in cooked and refrigerated meat, but not in the raw meat, whereas butanoic and nonanoic acids showed a decrease during the storage period. Finally, some authors [2] noticed that hexanoic acid was positively correlated with TBAR values and rancid odor intensity in beef steaks aged for 15 days under vacuum conditions, and displayed in modified atmosphere packaging during 9 days.

Pyrazines and sulfur compounds are important in cooked meat flavor due to the low odor detection threshold [9]. Statistical analysis showed that total nitrogen compounds decreased during the storage time. This outcome is in disagreement with data previously reported [10] highlighting an increase in some pyrazines (pyrazine, 2-methylpyrazine, 2,6-dimethylpyrazine, 2,3-dimethylpyrazine and 2-ethyl-6-methylpyrazine) in cooked beef, aged under vacuum conditions. According to these authors, the increase could be due to the release of free amino acids during the aging and high temperature, which enhances the Maillard reactions. In addition, it was previously recorded a significant increase of 2-methylpyrazine with increasing aging time in beef samples aged for 28 days [9]. Regarding sulfur compounds, only two (dimethyl sulfide and carbon disulfide) were detected in cooked foal meat samples aged for 14 days and they remained constant during all the investigated aging times. On the contrary, other authors observed a significant increase of methanethiol and dimethyl sulfide during the storage time of beef steaks aged for 28 days [9]. The increase of these sulfur compounds could be explained by the increase of sulfur-containing amino acids during the storage time. The VOCs formation, particularly for some compounds, showed an increasing trend until 6–9 days (depending by the compound) and sometimes a decreasing trend at 14 days. Volatile compounds analysis in muscle is made difficult for the low intramuscular fat content (often lower than 2%) and water distribution [51]. These aspects can probably lead to some differences when sampling for SPME extraction procedures. Moreover, although aldehydes are usually indicated as major contributors of meat flavor, and are the most present also in our samples, they can be converted to other volatile compounds during aging [52] and this could explain the production drop at 14 days compared to previous aging days.

Particularly interesting are results about enzymes activity in meat during aging. Usually, during aging, enzymes may lose their activity due to alterations within the muscle starting at slaughter, such as denaturation or hydrolysis of the enzyme by intracellular proteinases during storage, but also to redistribution between cellular compartments [53,54]. We observed that superoxide dismutase, catalase, and glutathione peroxidase increased their values during vacuum aging. These enzymes represent the in vivo primary mechanism for protecting cells from oxidative damage [55], and it is reported that in beef meat are relatively stable during refrigerated storage, offering some time post-mortem an antioxidant activity and a protection against free radical damage [56]. Aging leads also to reactive oxygen species (ROS) accumulation because of the oxidative and anti-oxidative balance breaking and the subsequent intracellular environment deterioration [57]. Different authors reported how these enzymes decreased during aging in beef [58] and lamb meat [42], reporting different results compared to ours. Renerre, et al. [59], studying aging effect on oxidative stability and antioxidant activity on turkey meat, reported results similar to ours. They observed that, after nine aging days overwrapped with an oxygen-permeable polyvinyl chloride (PVC) film, superoxide dismutase and glutathione reductase activity increased, but no explanation about these findings were reported. Our results are not able to state a real different behavior of oxidative enzyme profile during aging in horse meat, a hypothesis that could explain this trend is related to the unity of measurement of enzymes in meat. Particularly, vacuum aging and low film permeability induced different and slightly processes linked to oxidation and cellular degradation and a significant protein degradation, leading to a greater free amino acids availability. Because enzymes concentration is expressed as quantity on mg of proteins, the degradation of proteins could represent the reason of this increase, mainly base on a more rapid total protein degradation than enzymes, increasing the concentration, but not the total amount. Anyway, further studies are needed on horse muscle proteolytic degradation during vacuum aging.

Vacuum aging for 14 days affected some sensory parameters as juiciness, meaty odor, and overall licking. Aldehydes due to their low odor-detection threshold [37], had a huge impact on meat aromas [60], and their great presence in this meat (particularly Hexanal), can justify the high values recorded for meaty taste and overall licking and their increasing from the ninth aging day. Some ketones showed to change during aging, although they are produced in a little quantity, and probably could be able to affect meat aroma and so sensory evaluation [9]. It is probable that, considering the high threshold perception of some VOCs and the personal capacity of each one to perceive them, the decreasing trend that characterized some VOCs the last aging day was not able to affect sensory evaluation. Juiciness decreased, probably due to the protein degradation and a conceivable reduction in the meat ability to retain water, considering these two parameters [61]. The number of panelists is not high, and this may have influenced the final results concerning odor and overall licking during aging. Moreover, the limited number of samples included should be considered because this aspect may have affected the high VOCs production variability and, consequently, some particular VOC trends.

The overall outcomes from the present paper seems to be consistent with results reported in literature nowadays, increasing knowledge on the effect of vacuum aging in horse meat. The increase in volatile compounds during the first nine days of vacuum aging is consistent with the increase of consumer sensorial perception of intensity of flavor/aroma recently reported on vacuum aged horse meat during the same period [11]. The increase of volatile compounds can be linked to the chemical modifications of fat and protein molecules during storage [62].

## 5. Conclusions

The aging of foal meat under vacuum conditions for 14 days followed by cooking at 130–150 °C leads to small changes in the volatile compounds profile. The amount of total volatile compounds, as well as the main chemical categories of volatile compounds, tends to increase during storage, reaching a peak at 6–9 days from slaughtering. The oxidative profile deriving from lipid oxidation showed no variation during vacuum aging, confirming that these processes are slowed down by vacuum aging, probably due, also, to the low film permeability. It is possible that protein derivatives show how some non-oxidative processes are present and can influence both volatile compound production and sensory profile. Hexanal can be, probably, the most responsible of the sensory perception results, considering its high amount (probably over the perception threshold), although sensory results could be ascribed also to some rheological parameters that were not included in this study. The enzyme quantification showed an increasing trend that needs to be deeply investigated together with protein degradation and oxidation in horse meat during vacuum aging. These results contribute to the knowledge of vacuum aging techniques, showing that it can be used to maintain meat characteristics, but more investigation is needed.

## Figures and Tables

**Table 1 animals-10-01495-t001:** Effect of aging time in the aldehydes and furans content, expressed as quantifier area units (AU × 10^3^/g) of foal meat steaks aged for 14 days under vacuum conditions.

Volatile Compound	m/z	LRI	Aging Time (Days)				SEM	*p*-Value
			1	6	9	14		
Propanal	58	526	363.4	330.7	937.6	571.4	113.8	0.184
Propanal, 2-methyl-	72	557	62.4	87.8	51.1	62.6	10.0	0.610
Butanal	72	584	29.8	33.9	49.8	39.9	3.66	0.143
Butanal, 3-methyl-	58	659	55.9	77.0	184.9	92.2	32.0	0.360
Butanal, 2-methyl-	57	671	203.9	466.3	261.8	250.7	63.3	0.233
Pentanal	57	728	990.4	1152	1898	1418	250.6	0.567
Hexanal	56	865	16965	15704	27034	19657	3028	0.544
Furfural	95	933	25.0	17.3	21.6	22.7	2.20	0.527
2-Hexenal, (E)-	41	943	24.6 ^a^	23.9 ^a^	35.1 ^b^	26.4 ^a^	2.05	0.039
Heptanal	70	974	370.4 ^a^	317.1 ^a^	1420 ^b^	692.7 ^a^	153.9	0.013
Benzaldehyde	106	1045	112.2	162.9	334.6	198.3	33.1	0.054
5-Ethylcyclopent-1-enecarboxaldehyde	124	1099	26.2	21.7	29.0	27.2	1.79	0.306
Benzeneacetaldehyde	91	1119	35.6	31.8	240.4	91.0	47.7	0.306
2-Octenal, (E)-	70	1123	19.4	22.2	25.0	22.5	1.59	0.640
Nonanal	57	1148	289.9 ^a^	267.1 ^a^	744.0 ^b^	404.1 ^a,b^	70.9	0.014
Benzaldehyde, 3-ethyl-	134	1209	35.4	40.6	45.5	39.2	3.68	0.730
2,4-Decadienal	81	1315	16.3	16.3	14.4	15.9	0.42	0.105
**Total aldehydes**			**19713**	**18880**	**33782**	**23748**	**3609**	**0.419**
Furan, 3-methyl-	82	582	36.5	18.8	25.6	25.2	3.01	0.069
Furan, 2,5-dihydro-	41	670	18.0	18.8	26.0	20.7	1.38	0.116
Furan, 2-ethyl-	81	703	82.8 ^a^	98.7 ^a^	379.6 ^b^	179.0 ^c^	34.7	<0.001
2-n-Butyl furan	81	944	65.8 ^a^	71.1 ^a^	165.8 ^b^	91.7 ^a^	16.2	0.023
Furan, 2,3-dihydro-3-methyl-	81	984	18.0	18.6	55.9	39.1	11.5	0.373
Furan, 2-pentyl-	81	1043	588.0 ^a^	363.0 ^a^	4537.2 ^b^	1634 ^c^	482.5	<0.001
**Total furans**			**809.5** ^**a,b**^	**589.3** ^**b**^	**5190** ^**c**^	**1990** ^**a**^	**524.3**	**<0.001**

a–c Mean values in the same row (corresponding to the same parameter) not followed by a common letter differ significantly (*p* < 0.05; Tukey’s Test); SEM: standard error of mean; m/z: Quantifier ion; LRI: Lineal Retention Index calculated for DB-624 capillary column (J&W scientific: 30 m × 0.25 mm id, 1.4 μm film thickness) installed on a gas chromatograph equipped with a mass selective detector.

**Table 2 animals-10-01495-t002:** Effect of aging time in the hydrocarbons and aromatic hydrocarbons content, expressed as quantifier area units (AU × 10^3^/g) of foal meat steaks aged for 14 days under vacuum conditions.

Volatile Compound	m/z	LRI	Aging Time (Days)				SEM	*p*-Value
			1	6	9	14		
Pentane	43	500	20.2 ^a^	25.8 ^a^	45.0 ^b^	31.6 ^a,b^	3.26	0.012
n-Hexane	69	600	482.0	699.2	130.3	458.1	101.1	0.205
Hexane, 2,2-dimethyl-	57	660	184.9	267.9	97.2	171.4	25.3	0.107
Isopropylcyclobutane	56	670	20.4 ^a^	27.7 ^a,b^	39.4 ^b^	28.9 ^a,b^	2.34	0.010
Heptane	71	700	41.9	49.7	67.0	52.4	5.65	0.437
Pentane, 2,3,4-trimethyl-	71	756	61.1	69.1	28.9	53.1	7.26	0.209
Heptane, 3,3,4-trimethyl-	71	763	80.6	84.0	26.5	64.0	9.36	0.077
Hexane, 2,2,4-trimethyl-	57	804	122.8	103.2	42.4	95.5	14.4	0.135
2-Heptene, 3-methyl-	70	817	116.0	98.7	87.4	99.1	8.69	0.682
Octane	85	800	184.4	158.0	335.4	214.2	31.9	0.118
2-Octene, (E)-	112	833	10.2	10.2	10.1	10.1	0.01	0.493
Cyclohexane, 1,2-dimethyl- (cis/trans)	55	837	33.6 ^a^	27.6 ^a^	51.9 ^b^	35.1 ^a^	4.10	0.046
Heptane, 2,3-dimethyl-	43	847	43.5	30.0	22.6	36.5	4.34	0.061
4-Octene, (E)-	55	849	10.3	10.4	10.4	10.4	0.01	0.493
Bicyclo [2.2.2]octane	67	869	27.6	27.0	34.3	28.8	1.89	0.352
Octane, 2-methyl-	57	903	15.4	12.8	13.3	15.2	1.07	0.112
Heptane, 3-ethyl-	57	910	284.3	212.0	150.7	239.1	31.1	0.215
Nonane, 3,7-dimethyl-	57	920	142.4	83.4	86.1	128.6	20.8	0.117
Heptane, 2,2,4-trimethyl-	57	926	279.6	202.9	139.3	233.3	31.3	0.157
Heptane, 2-methyl-3-methylene-	57	935	33.4	27.0	23.9	32.0	3.94	0.261
Octane, 3,5-dimethyl-	57	940	229.1	189.0	127.1	201.2	25.4	0.224
3-Methyl-3-hexene	83	1042	52.0	51.0	38.9	50.0	5.50	0.628
2-Octene, 4-ethyl-, (E)-	69	982	136.6	136.1	102.8	138.5	18.7	0.497
Heptane, 3-ethyl-5-methylene-	70	989	292.9	290.3	187.9	267.6	30.6	0.478
3-Ethyl-3-methylheptane	57	992	81.3	86.8	55.6	79.1	9.21	0.433
Pentane, 3,3-dimethyl-	43	999	35.1	23.6	33.7	28.9	3.34	0.340
Undecane, 6,6-dimethyl-	57	1010	91.3	114.4	65.2	96.0	12.3	0.382
Nonane, 5-methylene-	56	1015	98.5	115.0	72.2	98.9	11.6	0.535
2-Nonene, 3-methyl-, (E)-	70	1026	239.7	295.4	202.3	251.7	28.8	0.685
Heptane, 2,2,4,6,6-pentamethyl-	57	1027	1887.8	2729.3	1650	2138	233.9	0.357
Decane	57	1000	386.9	619.6	309.5	437.6	58.0	0.291
(Z)-4-Methyl-2-hexene	98	1060	105.2	147.8	88.4	116.8	13.7	0.430
2,2,4,4-Tetramethyloctane	57	1066	193.9	327.0	153.0	229.0	29.3	0.156
Undecane, 5,5-dimethyl-	57	1084	154.6	228.9	126.0	175.0	20.0	0.256
Dodecane, 2,6,10-trimethyl-	57	1092	121.3	161.3	84.5	124.8	15.2	0.337
Dodecane, 4-methyl-	43	1105	82.2	118.0	69.8	92.6	10.3	0.346
2-Decene, 3-methyl-, (Z)-		1110	51.8	75.9	46.2	57.9	7.20	0.531
Undecane	57	1100	1048	1677	854.5	1196.3	141.9	0.201
2-Undecene, 9-methyl-, (Z)-		1137	220.8	338.8	189.9	251.8	29.4	0.315
2-Acetyl-2-methyltetrahydrofuran		1147	31.2	39.3	21.2	30.4	3.09	0.230
4,4-Dipropylheptane	57	1161	13.8	14.3	13.0	14.3	0.75	0.521
Pentane, 3,3-diethyl-	57	1166	40.2	63.3	95.3	62.9	12.4	0.415
Dodecane, 2-methyl-6-propyl-		1173	133.8	207.1	110.7	150.9	17.6	0.260
2-Undecene, 3-methyl-, (E)-	70	1181	35.4	59.1	34.1	42.2	5.21	0.344
Dodecane	57	1200	496.3	809.7	430.5	578.6	67.5	0.216
Pentadecane, 6-methyl-	57	1223	33.3	53.3	31.3	38.9	4.57	0.352
Decane, 2,3,6-trimethyl-		1238	16.8	16.7	18.1	17.1	0.68	0.864
Tridecane	71	1300	98.4	157.8	100.5	117.4	13.1	0.392
Tridecane, 3-methyl-	85	1304	14.8	18.2	15.4	15.8	0.94	0.619
Tetradecane	57	1328	13.9	13.9	13.4	13.8	0.09	0.129
**Total hydrocarbons**			**8509**	**11292**	**6678**	**9037**	**900.3**	**0.287**
Benzene	78	650	35.3 ^a^	38.2 ^a^	132.8 ^b^	66.6 ^c^	11.2	<0.001
Toluene	92	804	113.4	117.6	91.7	98.7	9.52	0.320
Ethylbenzene	91	917	139.7	430.0	162.3	224.5	58.8	0.324
Benzene, 1,3-dimethyl-	106	926	286.9	522.2	299.9	353.2	38.8	0.123
p-Xylene	106	958	68.2	132.5	72.6	88.7	10.7	0.139
3-Carene	136	983	99.0 ^a^	184.1 ^b^	76.1^a^	118.9 ^a,b^	14.3	0.032
**Total aromatic hydrocarbons**			**742.6 ^a^**	**1424.8 ^b^**	**835.7 ^a,c^**	**950.8 ^c^**	**143.5**	**<0.001**

a–c Mean values in the same row (corresponding to the same parameter) not followed by a common letter differ significantly (*p* < 0.05; Tukey’s Test); SEM: standard error of mean; m/z: Quantification ion; LRI: Lineal Retention Index calculated for DB-624 capillary column (J&W scientific: 30 m×0.25 mm id, 1.4 μm film thickness) installed on a gas chromatograph equipped with a mass selective detector.

**Table 3 animals-10-01495-t003:** Effect of aging time in the ketones and alcohols content, expressed as quantifier area units (AU × 10^3^/g) of foal meat steaks aged for 14 days under vacuum conditions.

Volatile Compound	m/z	LRI	Aging Time (Days)				SEM	*p*-Value
			1	6	9	14		
2,3-Butanedione	43	592	160.3	137.3	585.5	267.7	85.1	0.139
2-Butanone	72	596	189.0	212.8	86.1	181.1	31.1	0.252
2-Pentanone	86	720	19.2	14.4	16.6	16.6	0.79	0.183
3-Pentanone	57	736	74.9	37.5	188.8	102.3	35.7	0.507
2,3-Pentanedione	100	739	81.7	83.4	143.7	111.9	14.7	0.272
Acetoin	45	787	146.1	123.5	569.1	278.8	1067.5	0.152
3-Heptanone	57	960	16.2 ^a^	23.1 ^b^	27.6 ^b^	20.3 ^c^	1.92	0.022
2-Heptanone	58	967	144.5 ^a^	103.7 ^a^	905.2 ^b^	347.6 ^c^	91.4	<0.001
4-Cyclopentene-1,3-dione	96	1003	33.3	42.0	31.9	34.2	2.63	0.486
4-Hexen-3-one, 5-methyl-	83	1047	23.6 ^a^	18.1 ^a^	35.1 ^b^	24.0 ^a^	2.58	0.022
Butyrolactone	42	1049	131.1	174.9	93.6	144.3	16.3	0.121
2(5H)-Furanone	55	1053	102.2	47.4	308.6	167.2	48.3	0.238
3-Octen-2-one	55	1111	27.4 ^a^	27.4 ^a^	22.0 ^b^	26.6 ^a^	1.14	0.045
3-Octanone, 2-methyl-	43	1141	27.6	31.8	29.9	29.6	1.46	0.828
2-Nonanone, 3-(hydroxymethyl)-	43	1146	14.2	12.5	14.6	14.0	0.53	0.568
**Total ketones**			**1191**	**1090**	**3058**	**1766**	**1229**	**0.066**
Cyclobutanol	44	504	147.5	130.0	431.1	257.7	52.0	0.107
1-Pentanol	55	847	603.6 ^a^	631.8 ^a^	3253 ^b^	1332 ^c^	343.6	<0.001
1-Hexanol	56	959	865.0	2135	10872	3974	2081.9	0.242
1-Heptanol	70	1046	88.7	72.4	262.1	124.7	36.1	0.117
1-Octen-3-ol	57	1051	211.5 ^a^	153.6 ^a^	757.5 ^b^	347.3 ^c^	73.5	<0.001
n-Tridecan-1-ol	55	1073	110.9	155.6	207.1	153.4	18.3	0.270
1-Heptanol, 2,4-diethyl-	69	1085	140.6	199.3	66.1	143.6	21.0	0.081
1-Decanol	70	1027	18.1	21.1	13.6	18.3	1.64	0.281
1-Tetradecanol	68	1225	23.8	29.0	15.4	23.2	2.16	0.118
1-Decanol, 2-hexyl-	97	1241	17.8	19.7	16.3	18.6	1.23	0.576
1-Butanol, 2-methyl-	57	1243	15.2	13.7	13.2	14.7	0.61	0.188
1-Octanol, 2-methyl-	57	177	17.0	18.3	13.8	17.5	1.49	0.306
**Total alcohols**			**2260 ^a^**	**3580 ^a^**	**15922 ^b^**	**6426 ^a^**	**2535**	**0.048**

a–c Mean values in the same row (corresponding to the same parameter) not followed by a common letter differ significantly (*p* < 0.05; Tukey’s Test); SEM: standard error of mean; m/z: Quantification ion; LRI: Lineal Retention Index calculated for DB-624 capillary column (J&W scientific: 30 m × 0.25 mm id, 1.4 μm film thickness) installed on a gas chromatograph equipped with a mass selective detector.

**Table 4 animals-10-01495-t004:** Effect of aging time in the carboxylic acids, total nitrogen, and sulfur compounds content, expressed as quantifier area units (AU × 10^3^/g) of foal meat steaks aged for 14 days under vacuum conditions.

Volatile Compound	m/z	LRI	Aging Time (days)				SEM	*p*-Value
			1	6	9	14		
Butanoic acid	60	918	95.2 ^a^	310.4 ^b^	174.4 ^c^	178.9 ^c^	30.7	0.046
Hexanoic acid	60	1088	72.3	61.7	135.7	83.8	18.6	0.408
Formic acid, octylester	56	1133	83.2	41.7	79.0	63.7	9.92	0.291
**Total carboxylic acids**			**178.5** ^**a**^	**413.8** ^**b**^	**389.2** ^**b**^	**271.5** ^**c**^	**48.9**	**0.045**
Diazene, dimethyl-	15	532	377.3	379.6	229.3	354.9	33.6	0.085
Pyrazine, methyl-	94	893	146.5 ^a^	95.4 ^a^	62.5 ^b^	96.0 ^a^	10.5	0.005
2-Propen-1-amine	56	916	115.0	64.6	55.8	91.2	17.1	0.336
Pyrazine, 2,5-dimethyl-	42	982	413.1	273.7	154.1	253.4	43.5	0.082
Pyrazine, trimethyl-	122	1064	264.7 ^a^	120.2 ^b^	65.5 ^c^	149.4 ^b^	22.5	0.001
**Total nitrogen compounds**			**1316.8** ^**a**^	**933.6** ^**b**^	**567.3** ^**c**^	**945.1** ^**b**^	**87.7**	**0.004**
Dimethylsulfide	62	534	21.5	24.4	17.3	25.2	3.70	0.133
Carbon disulfide	76	533	111.9	73.4	127.9	113.2	13.6	0.450
**Total sulfur compounds**			**136.3**	**94.7**	**106.2**	**114.7**	**10.9**	**0.607**

a–c Mean values in the same row (corresponding to the same parameter) not followed by a common letter differ significantly (*p* < 0.05; Tukey’s Test); SEM: standard error of mean; m/z: Quantification ion; LRI: Lineal Retention Index calculated for DB-624 capillary column (J&W scientific: 30 m × 0.25 mm id, 1.4 μm film thickness) installed on a gas chromatograph equipped with a mass selective detector.

**Table 5 animals-10-01495-t005:** Effect of aging time on oxidative profile (thiobarbituric acid reactive substances ((TBARS), hydroperoxides, and protein carbonyls), and antioxidant enzymes activity (superoxide dismutase, catalase, and glutathione peroxidase of foal meat steaks aged for 14 days under vacuum conditions.

Item	Aging Time (Days)				SEM	*p*-Value
	1	6	9	14		
Oxidative profile						
TBARS (mg MDA/kg of meat)	0.69	0.67	1.05	0.99	0.10	0.5351
Hydroperoxides (mmol/g of meat)	0.48	0.44	0.53	0.41	0.05	0.8125
Protein carbonyls (mmol DNPH/mg protein)	1.42 ^A^	1.62 ^A^	1.72 ^A^	2.94 ^B^	0.14	0.0004
**Antioxidant enzymes**						
Superoxide dismutase (U/mg protein)	13.23 ^A^	17.33 ^B^	21.38 ^C^	24.45 ^D^	0.15	<0.001
Catalase (U/mg protein)	3.56 ^A^	4.26 ^B^	5.11 ^C^	5.97 ^D^	0.05	<0.001
Glutathione peroxidase (µmol NADPH ox/mg protein)	8.52 ^A^	9.77 ^B^	10.99 ^C^	12.17 ^D^	0.05	<0.001

A–D Mean values in the same row (corresponding to the same parameter) not followed by a common letter differ significantly (*p* < 0.01; Tukey’s Test); SEM: standard error of mean, MDA: malondialdehyde; DNPH: 2,4-dinitrophenylhydrazine; NADPH: nicotinamide adenine dinucleotide phosphate reductase

**Table 6 animals-10-01495-t006:** Effect of aging time on sensory panel evaluation foal meat steaks aged for 14 days under vacuum conditions.

Item	Aging Time (Days)				SEM	*p*-Value
	1	6	9	14		
Tenderness	6.02	6.08	6.18	6.15	0.06	0.5121
Juiciness	6.12	6..22 ^a^	5.59 ^b^	5.51 ^b^	0.08	0.0242
Sweetness	6.78	6.65	6.80	6.76	0.12	0.1548
Unpleasant taste	5.54	5.74	5.68	5.47	0.42	0.4120
Unpleasant odor	6.21	6.12	6.05	6.22	0.12	0.2771
Meaty odor	7.04 ^a^	7.01 ^a^	7.41 ^b^	7.40 ^b^	0.15	0.0155
Overall liking	6.88 ^a^	6.76 ^a^	7.21 ^b^	7.24 ^b^	0.09	0.0207

a–b Mean values in the same row (corresponding to the same parameter) not followed by a common letter differ significantly (*p* < 0.05; Tukey’s Test); SEM: standard error of mean.

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
