# Peer review of "Volatile Organic Compounds, Oxidative and Sensory Patterns of Vacuum Aged Foal Meat"

_animals, 2020, doi:10.3390/ani10091495_

Round 1

Reviewer 1 Report

The authors reported a study on the effect of 14-days vacuum aging on the volatile compounds profile, oxidative profile, antioxidant enzymes activity and sensory evaluation in Longissimus thoracis muscle of foal meat under vacuum aging. However, it is not clear how many samples were analyzed for each test at each time point, i.e., each aging time day? Please add this information in the Materials and Methods part.

Line 29: “TBARs”, please give the full name of it.

Author Response

Dear Editor,

Please find attached a new version of our paper entitled “Volatile organic compounds, oxidative and sensory patterns of vacuum aged foal meat” - that was submitted to Animals for publication.

The manuscript has been carefully rechecked, and appropriate changes have been made in accordance with the reviewers’ suggestions. We would like to thank the referees for their excellent comments. They have helped improve the manuscript quality significantly, and we believe that the manuscript now provides a more balanced and better account of the research. We have modified the manuscript accordingly, and the detailed corrections are listed below point by point. Changes to the manuscript have been added on the manuscript with the “Tracked changes” option active.

Please do not hesitate to contact me again if further changes to the manuscript are required.

Sincerely,

Aristide Maggiolino

Reviewer 1

The authors reported a study on the effect of 14-days vacuum aging on the volatile compounds profile, oxidative profile, antioxidant enzymes activity and sensory evaluation in Longissimus thoracis muscle of foal meat under vacuum aging. However, it is not clear how many samples were analyzed for each test at each time point, i.e., each aging time day? Please add this information in the Materials and Methods part.

Au: thank you for your observation, it is important to improve the clarity of the M&M section. In lines 98-99 now we specify that we collected 10 samples for each aging time. Moreover, for each analysis, is now specified how many samples we collected (line 105 for VOCs, 139 for TBARs and hydroperoxides, 159 for protein carbonyls, 174 for enzymes and 202-203 for sensory analysis.

Line 29: “TBARs”, please give the full name of it.

Au: done

Reviewer 2 Report

This paper describes the volatiles found in cooked foal meat after various lengths of storage in vacuum packaging. It will be of interest to flavour chemists but is limited in scope as many countries do not eat horse meat and the length of storage period is limited.

More specifically:  The English needs some attention and the references need to be more precise.

Introduction.

Reference 1 is not adequate as an overall reference for meat ageing, it is too specific. Any older review, especially any involving Koohmaraie. Reference 2 is not about ageing in vacuum but in High oxygen retail packs.

Where does 7-21 days ageing come from? Beef is often aged to 35 days, there is a recent paper (Meat Science) looking at much longer storage periods. You might also look at other older flavour reviews e.g. Mottram, Falmer. Reference 4, specific to foal, already known that high temperatures required for maillard browning and flavour development occur at meat surfaces during grilling or frying.

Line 63. Need to be careful using content.  There is a difference between concentration and proportions. There may be greater proportion of PUFA in horsemeat, but because it is leaner there are probably lower concentrations of SATs and MUFAs. Healthier because low fat, but does it contain a high enough concentration of long chain, n-3 PUFA? (see EU regulations on labelling of healthy products.

L 77. Why this sentence saying high oxygen leads rancid flavour when not explained or apparently relevant to this paper.

Methods.

Line 96-7. What was the permeability of the vacuum bags?

L 103. Steaks were grilled at 130-150

L137. Cooked or raw meat was minced l156, cooked or raw.

L147. Is this the hydroperoxides method? Not clear and not in the order of the subheading

L171, raw meat

L181 H2O2, 2 as subscripts

L184-187, GSSG and GR not defined.

L198. What was a session. It is recommended that panellists do not assess more than 6 samples at a sitting. Four panels with a break between each would allow 24 samples in a morning. Overall liking is a hedonic measurement. Trained panels of 8 people are not representative of the population, so you should make some statement that effect and suggest that this parallels the increase in odour.

L207, doesn’t mention 3 days which was stated l96

Were the results for TBARS tested for linearity. Development over time is exponential and would thus be log transformed before statistical analysis.

Table 6. I am surprised that such small differences in sensory attributes are so statistically significantly different. Was n used 10 or 20?

L294-295. Most authors find that volatiles increase with storage time. You do not discuss why yours increased and then decreased.

Line 314. The foals were all on the same diet so meat composition should have been similar. How variable was the fat content? Was there any relationship between fat concentration or concentrations of individual volatiles and the sensory results?

L353. Other studies show that -S-S- bonds increase over time, but usually in high oxygen and toughen the meat. In vacuum oxidation should not happen. Over time, depending upon the permeability of the vacuum pouches, oxygen permeates into the bag, can no-longer be used up by the now declining enzymes (and co-factors) in the meat surface. Myoglobin will oxidise, the meat discolours and rancid odour/flavour/TBARS develop. What was the permeability of your vacuum bags?

L436. You homogenised and sampled a thick sample. All the oxidative reactions occur at the meat surface so these protein changes would have been relatively small. What activates the antioxidant enzymes?

The changes in sensory preference, increasing with time, do not parallel the volatiles which rise and then fall. This not discussed.

L460-461. How was it shown?

Author Response

Dear Editor,

Please find attached a new version of our paper entitled “Volatile organic compounds, oxidative and sensory patterns of vacuum aged foal meat” - that was submitted to Animals for publication.

The manuscript has been carefully rechecked, and appropriate changes have been made in accordance with the reviewers’ suggestions. We would like to thank the referees for their excellent comments. They have helped improve the manuscript quality significantly, and we believe that the manuscript now provides a more balanced and better account of the research. We have modified the manuscript accordingly, and the detailed corrections are listed below point by point. Changes to the manuscript have been added on the manuscript with the “Tracked changes” option active.

Please do not hesitate to contact me again if further changes to the manuscript are required.

Sincerely,

Aristide Maggiolino

Reviewer 2

This paper describes the volatiles found in cooked foal meat after various lengths of storage in vacuum packaging. It will be of interest to flavour chemists but is limited in scope as many countries do not eat horse meat and the length of storage period is limited.

Au: Yes, we know that many countries don’t eat horsemeat, however Italy, as many other European and not European countries traditionally eat this meat. Although this low diffusion of horse meat, as for other meat products we are interested to continue our research.  Moreover, the EU regulation on horses for food production and horses for other purposes guarantees higher safety of horse meat in comparison with other species. In addition, both published and unpublished data from our research team, as well as from other ones highlights that shelf life of horse meat is not different from meat from other species. Besides, horse meat production guarantees also agricultural biodiversity, as many cold-blood breeds used in the past for work, now risk endangerment if they are not used for meat. Finally, several papers dealing with nutritional properties of horse meat highlight the potential role of this food on human health impact.  Because horses are herbivores but not ruminants we suppose that the environmental impact for producing 1kg of horse meat is lower than beef, pork, lamb…, although on this topic there is the need of further researchers. All this strengthens  our idea of continuing to investigate on this field. 

More specifically:  The English needs some attention and the references need to be more precise.

Introduction.

Reference 1 is not adequate as an overall reference for meat ageing, it is too specific. Any older review, especially any involving Koohmaraie. Reference 2 is not about ageing in vacuum but in High oxygen retail packs.

Au: reference 1 was changed, as well as the sentence. We know that reference 2 is not about ageing in vacuum, but it was not our will to refer that sentence to vacuum ageing, but to ageing in general (as reported also in cited paper). For this reason, we think to maintain that sentence with that reference.

Where does 7-21 days ageing come from? Beef is often aged to 35 days, there is a recent paper (Meat Science) looking at much longer storage periods. You might also look at other older flavour reviews e.g. Mottram, Falmer. Reference 4, specific to foal, already known that high temperatures required for maillard browning and flavour development occur at meat surfaces during grilling or frying.

Au: yes, we agree with beef storage time. It was deleted and the sentence revised. Reference 4 is specific to foal, as the sentence before. We think it is correct and our will is to maintain it. 

Line 63. Need to be careful using content.  There is a difference between concentration and proportions. There may be greater proportion of PUFA in horsemeat, but because it is leaner there are probably lower concentrations of SATs and MUFAs. Healthier because low fat, but does it contain a high enough concentration of long chain, n-3 PUFA? (see EU regulations on labelling of healthy products.

Au: thank you for this suggestion, it is very important for clarity and scientific soundness of our paper. We corrected it.

L 77. Why this sentence saying high oxygen leads rancid flavour when not explained or apparently relevant to this paper.

Au: the sentence was deleted.

Methods.

Line 96-7. What was the permeability of the vacuum bags?

Au: Sorry for the lack. Now film characteristics are reported in M&M section.

L 103. Steaks were grilled at 130-150

Au: done

 L137. Cooked or raw meat was minced l156, cooked or raw.

Au: now it is specified. Thanks you for suggestion

L147. Is this the hydroperoxides method? Not clear and not in the order of the subheading

Au: sorry, it was revised to clarify.

L171, raw meat

Au: done

L181 H2O2, 2 as subscripts

Au: done

L184-187, GSSG and GR not defined.

Au: done

L198. What was a session. It is recommended that panellists do not assess more than 6 samples at a sitting. Four panels with a break between each would allow 24 samples in a morning. Overall liking is a hedonic measurement. Trained panels of 8 people are not representative of the population, so you should make some statement that effect and suggest that this parallels the increase in odour.

Au: sorry for the lack of clarity. The sentence was revised to better clarify. For each test day 4 sitting were done of 3 or 2 different steaks (respectively 6 or 4 different samples). Moreover, considering that we had only 8 panellists (a number that is similar, and sometimes higher than other papers) we now specify that this can be also the main cause of some differences in our results.

L207, doesn’t mention 3 days which was stated l96

Au: Now it is specified

Were the results for TBARS tested for linearity. Development over time is exponential and would thus be log transformed before statistical analysis.

Au: Yes, all data were subjected to test of normality (Shapiro- Wilk)

Table 6. I am surprised that such small differences in sensory attributes are so statistically significantly different. Was n used 10 or 20?

Au: we used n=10 samples (the mean value for the double sensory test for each sample). We agree with your surprise, however the significance is P < 0.05, not higher.  

L294-295. Most authors find that volatiles increase with storage time. You do not discuss why yours increased and then decreased.

Au: we agree, the trend of some VOCs is particular, considering that they tend to increase until 9 days and decreased at 14 days. Some others tend to remain constant or to increase until 14 days with no statistical differences. AS we say the high variability of the VOCs values can lead also to some differences or some lack of statistical differences. However, the expression of VOCs as percentage of the total can lead some singular VOCs to be less represented because of the increasing of others, although these last didn’t showed statistical differences. Moreover, as we now underline, the low film permeability lead the VOCs trend to be unusual probably due to the slowed down or absent lipid oxidation processes that usually characterized meat ageing and VOC’s production.

Line 314. The foals were all on the same diet so meat composition should have been similar. How variable was the fat content? Was there any relationship between fat concentration or concentrations of individual volatiles and the sensory results?

Au: The chemical analysis of meat was performed only at day 1 and no differences were observed between horses, considering that these animals had same diet and same age at slaughter. There is not a great variability between fat concentration, and the sensory valuation can be linked to individual compounds and individual capacity of perception of the same compounds. However, it is now reported this aspect in the paper. 

L353. Other studies show that -S-S- bonds increase over time, but usually in high oxygen and toughen the meat. In vacuum oxidation should not happen. Over time, depending upon the permeability of the vacuum pouches, oxygen permeates into the bag, can no-longer be used up by the now declining enzymes (and co-factors) in the meat surface. Myoglobin will oxidise, the meat discolours and rancid odour/flavour/TBARS develop. What was the permeability of your vacuum bags?

Au: We totally agree with you. Probably the poor permeability of film used (now reported in the M&M section) linked to the short time considered affected the VOCs formation and the lipid oxidation. This aspect is important for future studies about vacuum packaging, films and storage time.  

L436. You homogenised and sampled a thick sample. All the oxidative reactions occur at the meat surface so these protein changes would have been relatively small. What activates the antioxidant enzymes?

Au: We agree, all the oxidative processes started from the surface and with time they go deeply. However, in this case we refer to protein degradation not necessary linked to oxidation processes. Moreover, we can’t say that enzymes are more active and/or they have some activation processes due to oxidative processes on meat. Although oxidation seems to be blocked, it probably slightly happens on meat, but we can’t say that these processes cause enzymes activation. In fact, we suppose that our results are probably due to a greater enzymes concentration for the substrate degradation.

The changes in sensory preference, increasing with time, do not parallel the volatiles which rise and then fall. This not discussed.

L460-461. How was it shown?

Au: the sentence was revised

Reviewer 3 Report

Dear Authors 

It is a very interesting paper dealing with aging of foal meat and several compounds, determinants of sensorial properties.

However some simple questions should be addressed. Please note that English is not my native language.

Line 16 - "Aging is..." in my opinion it is oversimplified. Plesase explain better and rephrase or explain it better in "Introduction".

Line 110 - "SPME extraction". Please explain in full the meaning of SPME.

Lines 215-216 - What is the difference in the evolution of the values of Heptanal, 2-Hexenal and nonanal that justifies two separated sentences? Why didn't you mention all these compounds in one sentence? 

Table 1 – In my opinion,  in the title, it must be “aldehydes and furans” following the same order that these compounds are described in the table.  

Line 240 – in my opinion, 3-heptatnone increase until 9 days although there are no significant diferences from 6 days. Please confirm.

Table 4 – in the title, it should be “total nitrogen” instead “nitrogen”

Tables 5 and 6 – What does it mean IEM? Please explain.

Author Response

Dear Editor,

Please find attached a new version of our paper entitled “Volatile organic compounds, oxidative and sensory patterns of vacuum aged foal meat” - that was submitted to Animals for publication.

The manuscript has been carefully rechecked, and appropriate changes have been made in accordance with the reviewers’ suggestions. We would like to thank the referees for their excellent comments. They have helped improve the manuscript quality significantly, and we believe that the manuscript now provides a more balanced and better account of the research. We have modified the manuscript accordingly, and the detailed corrections are listed below point by point. Changes to the manuscript have been added on the manuscript with the “Tracked changes” option active.

Please do not hesitate to contact me again if further changes to the manuscript are required.

Sincerely,

Aristide Maggiolino

Reviewer 3

Dear Authors 

It is a very interesting paper dealing with aging of foal meat and several compounds, determinants of sensorial properties.

However some simple questions should be addressed. Please note that English is not my native language.

Line 16 - "Aging is..." in my opinion it is oversimplified. Plesase explain better and rephrase or explain it better in "Introduction".

Au: It was revised and rephrased to better explain, Thank you for your suggestion

Line 110 - "SPME extraction". Please explain in full the meaning of SPME.

AU: it is now explained

Lines 215-216 - What is the difference in the evolution of the values of Heptanal, 2-Hexenal and nonanal that justifies two separated sentences? Why didn't you mention all these compounds in one sentence? 

AU: Thank you for this suggestion. Now it is written in a single sentence.

Table 1 – In my opinion,  in the title, it must be “aldehydes and furans” following the same order that these compounds are described in the table.  

AU: done

Line 240 – in my opinion, 3-heptatnone increase until 9 days although there are no significant diferences from 6 days. Please confirm.

AU: the sentence was revised to better clarify

Table 4 – in the title, it should be “total nitrogen” instead “nitrogen”

AU: done

Tables 5 and 6 – What does it mean IEM? Please explain.

AU: sorry, there was a typing error. The word is “Item” and it is now correct.

Round 2

Reviewer 1 Report

The authors has clarified the questions raised in the first review and updated the manuscript. While, the concern to me now is that 10 samples at each time point is really a small sample size, which means it is unsure whether the 10 samples can give a very robust result. So I would strongly suggest the authors to add explanations in the Conclusion part or somewhere appropriate to clarify why this small sample size was used in this study and why not use more samples. 

Author Response

The authors has clarified the questions raised in the first review and updated the manuscript. While, the concern to me now is that 10 samples at each time point is really a small sample size, which means it is unsure whether the 10 samples can give a very robust result. So I would strongly suggest the authors to add explanations in the Conclusion part or somewhere appropriate to clarify why this small sample size was used in this study and why not use more samples. 

Au: Thank you for your remark. We understand concerns about number of samples, only 10. WE want to explain why this “limited” number. Horse meat is not worldwide diffused, and there are not a lot of animals reared in the same farm at the same condition. Considering this, it is not easy to obtain the same product by the same breed (and breeder) so to evaluate differences that can be ascribed to aging time and packaging. Often, papers relating to horse meat had a “small” samples number (eight, ten, twelve…) because of these aspects. However, we agree that the number is not high, and we now underline this aspect in the paper.

Reviewer 2 Report

Many of the points raised have now been considered. The English still needs some attention, which is presumably done by the journal.

As an example.

At each aging time, two 5g samples of each foal steak were grilled using a grilled at 130-150 ºC  for 5 min on each surface, using an electrical griddle (Delonghi ......

Still no theory as to why volatiles increase and then decrease. Are they converted further into other compounds?  Much of the change would have been on the surface. Was the same sample cut from the aged meat each time so that they had the same amount of newly exposed and old, oxidised surface?

Which of your volatiles, or combinations of volatiles might be responsible for flavour changes.

Author Response

Many of the points raised have now been considered. The English still needs some attention, which is presumably done by the journal.

As an example.

At each aging time, two 5g samples of each foal steak were grilled using a grilled at 130-150 ºC  for 5 min on each surface, using an electrical griddle (Delonghi ......

Au: we are sorry, it is now corrected. It was probably a typing error

Still no theory as to why volatiles increase and then decrease. Are they converted further into other compounds?  Much of the change would have been on the surface. Was the same sample cut from the aged meat each time so that they had the same amount of newly exposed and old, oxidised surface?

Au: Thank you for your suggestion. No, it was not the same sample. Each sample was packaged singularly and randomly assigned to the aging time. Although it was the same muscle sampled in a limited area of the muscle itself, they were different and new samples for each aging time. There was not old oxidation or any other previous activity on it. However, we now added some possible theories on VOCs trend.

Which of your volatiles, or combinations of volatiles might be responsible for flavour changes

Au: Now we add some hypothesis. Thank you for your suggestion